# A simple retinal mechanism contributes to perceptual interactions between rod- and cone-mediated responses in primates

**William N Grimes, Logan R Graves[†], Mathew T Summers[†], Fred Rieke***

Department of Physiology and Biophysics, Howard Hughes Medical Institute, University of Washington, Seattle, United States

**Abstract** Visual perception across a broad range of light levels is shaped by interactions between rod- and cone-mediated signals. Because responses of retinal ganglion cells, the output cells of the retina, depend on signals from both rod and cone photoreceptors, interactions occurring in retinal circuits provide an opportunity to link the mechanistic operation of parallel pathways and perception. Here we show that rod- and cone-mediated responses interact nonlinearly to control the responses of primate retinal ganglion cells; these nonlinear interactions, surprisingly, were asymmetric, with rod responses strongly suppressing subsequent cone responses but not vice-versa. Human psychophysical experiments revealed a similar perceptual asymmetry. Nonlinear interactions in the retinal output cells were well-predicted by linear summation of kinetically-distinct rod- and cone-mediated signals followed by a synaptic nonlinearity. These experiments thus reveal how a simple mechanism controlling interactions between parallel pathways shapes circuit output and perception.

*For correspondence: rieke@u.washington.edu

[†]These authors contributed equally to this work

**Competing interests:** The authors declare that no competing interests exist.

## Introduction

Perceptual interactions between rod- and cone-mediated signals have been studied for nearly 200 years; these interactions influence the spatial, temporal and chromatic sensitivities of human vision at light levels ranging from moonlight to dawn or dusk (for review see *Buck, 2004, 2014*). Despite the impact of rod-cone interactions on vision, little is known about the mechanistic basis of such interactions. Our aims here were to relate perceptual rod-cone interactions to retinal mechanisms and by doing so to provide an example of how the mechanisms controlling parallel processing in neural circuits impact computation and human perception.

## Results

We first compared rod-cone interactions in the output signals of the primate retina with those observed perceptually (*Figure 1*). We focused on nonlinear interactions between brief increment flashes that preferentially elicited responses from ON retinal circuits; ganglion cell spike outputs in response to these flashes were dominated by excitatory synaptic input (*Figure 1—figure supplement 1*), further simplifying the circuitry involved. ON parasol ganglion cells (which project to magnocellular layers of the lateral geniculate nucleus) exhibited particularly robust responses to such flashes. We used dim short-wavelength flashes to preferentially activate rod photoreceptors, and brighter long-wavelength flashes to preferentially activate L-cone photoreceptors (*Figure 1A,B*; *Figure 1—figure supplement 2*).

The linear sum of ON parasol responses to rod- and cone-preferring flashes delivered individually differed substantially from the response to the flashes delivered together (*Gouras and Link, 1966*) (*Figure 1B*)—that is, rod- and cone-mediated signals interact nonlinearly within the retina. We quantified the strength of the nonlinear interactions using an interaction index (see 'Materials and methods');

**eLife digest** The inner surface at the back of the eye is called the retina and contains two types of light-sensitive cells: rod cells and cone cells. Rods outnumber cones by roughly twenty to one and are responsible for vision under low light levels. Cone cells, by contrast, provide detailed vision in bright light, as well as the ability to see in color.

Rods and cones provide input to two distinct networks of cells that convey information in parallel to cells called ganglion cells, which then relay this information out of the retina. However, the signals from activated rods can feed into the cone pathway at several points, meaning that the responses of rods and cones are not independent. At dawn and dusk—and indeed under street lighting at night—rods and cones are both active and interactions between rod and cone responses influence many aspects of vision, including sensitivity to color and contrast.

Grimes et al. have now identified a neural mechanism behind these interactions by combining measurements of human vision with recordings of electrical activity in retinas from non-human primates. The experiments confirmed that activating either type of photoreceptor briefly suppresses the responses of the other, although unexpectedly rods inhibit cones more than cones inhibit rods. The site of this interaction is the connection—or synapse—between the very last cell in the cone pathway and the retinal output cells. Prior to this 'gateway' synapse, rod and cone-mediated responses are largely independent.

Vision at dawn and dusk is shaped by a complex set of interactions between rod and cone signals—such as the ability of activated rods to change color perception at dusk. These findings show that these seemingly complex behaviors can arise from simple interactions at the level of neural circuits.

an interaction index of 0 corresponds to linear summation of the responses, whereas an interaction index of 1 indicates that the 'adapt' flash (i.e., the first flash in the sequence) completely suppressed the response to the 'test' flash (i.e., the second flash in the sequence).

Switching the identity of the adapt and test flashes revealed a surprising asymmetry in these interactions: cone responses produced a modest suppression of subsequent rod responses (cone → rod interaction), while rod responses produced a strong suppression of subsequent cone responses (rod → cone interaction; *Figure 1C,D*). This asymmetry held for rod- and cone-preferring adapt flashes that produced similar amplitude responses in the RGC (as in *Figure 1C*). Rod → cone interactions were restricted to a <1 s time window following the adapt flash (*Figure 1E,F*). ON midget ganglion cells, another prominent retinal ganglion cell type in primate retina, exhibited qualitatively similar rod-cone interactions (data not shown).

Do asymmetric rod-cone interactions occur perceptually? To answer this question we asked human observers to match the perceived brightness of two test flashes—one at the same location as the adapt flash and the other displaced spatially (*Figure 1G*). Comparing matches from trials in which the adapt flash was located in the same or the opposite eye as the test flashes allowed us to separate interactions that likely originated in the retina (monocular-specific interactions) from those that likely originated in the cortex (binocular interactions; see 'Materials and methods' and *Figure 1—figure supplement 3*). Monocular-specific interactions shared several features with those observed in the responses of ON parasol ganglion cells, although several issues, including uncertainty about how retinal and cortical interactions combine, precluded quantitative comparison: (1) rod → cone interactions were stronger than cone → rod interactions (*Figure 1H*); and, (2) rod → cone interactions were stronger for 0.2–0.3 s intervals between adapt and test flashes than for 1 s intervals (*Figure 1I*). These similarities suggest that retinal rod-cone interactions contribute substantially to perceptual interactions; this similarity motivated investigation of where and how the retinal interactions occur.

Previous studies provide clear, testable predictions for where rod → cone interactions might originate (*Kolb, 1977*, *1979*; *Tsukamoto et al., 2001*; *Field et al., 2009*). Rod-mediated signals can traverse the retina through the dedicated rod bipolar circuitry and/or through cones via rod-cone gap junctions and then through the associated cone bipolar circuitry (*DeVries and Baylor, 1995*; *Schneeweis and Schnapf, 1995*) (*Figure 1A*). To determine which route dominates under our experimental conditions, we compared RGC sensitivity to rod- and cone-preferring flashes with the

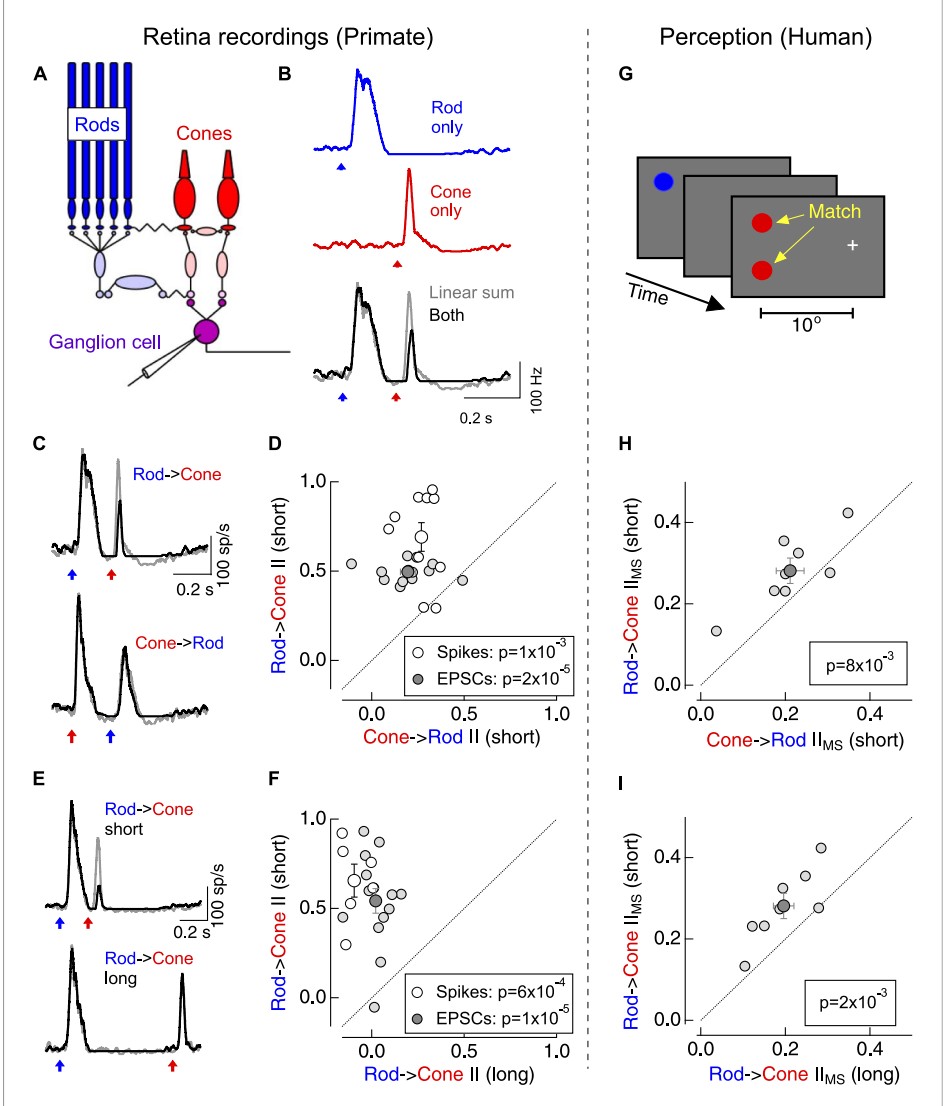

**Figure 1**. Asymmetric nonlinear rod-cone interactions. (**A**) Diagram of the retinal circuits that convey rod- and cone-generated signals to the brain. Dim short-wavelength light preferentially activates rod photoreceptors whereas long-wavelength light preferentially activates long wavelength (L) cone photoreceptors (see *Figure 1—figure supplement 2*). (**B**) Protocol for testing for nonlinear rod-cone interactions electrophys-iologically. Neural responses in ON parasol ganglion cells to rod (top) and cone (middle) flashes were recorded separately, and the sum of these responses (bottom, gray) was compared to trials in which the rod and cone flashes were delivered together (bottom, black). (**C**) Average spike responses from an example cell comparing rod → cone (top) and cone → rod (bottom) interactions. (**D**) Summary data across cells comparing rod → cone and cone → rod interaction indices for spikes and excitatory postsynaptic currents (EPSCs). (**E**) Average spike responses from an example cell comparing rod → cone interactions for short (0.2 s, top) and long (0.8 s, bottom) time offsets. (**F**) Summary data across cells: short intervals were 0.2–0.3 s and long intervals were ≥0.8 s. (**G**) Protocol used to test for nonlinear rod-cone interactions in human perception (also see 'Materials and methods' and *Figure 1—figure supplement 3*). Observers fixated on a cross while rod and cone flash sequences (similar to electrophysiology stimuli) were delivered to their peripheral retina (~10° eccentricity). Observers compared the perceived brightness of two test flashes, one in the same location as the initial flash and the other spatially offset. (**H**) Comparison of perceptual rod → cone and cone → rod interactions for 8 human observers. (**I**) Comparison of perceptual rod → cone interactions for short and long time offsets. Each marker in panels **D** and **F** represents the average interaction for a single ganglion cell. Each marker in **H** and **I** represents the average monocular pathway-specific interaction for a single observer

*Figure 1. continued on next page*

*Figure 1. Continued*

(see 'Materials and methods' and *Figure 1—figure supplement 3*). All retinal recordings (**B**–**F**) from whole mount retina.

The following figure supplements are available for figure 1:

**Figure supplement 1**. Rod → cone interactions in RGC spike output are largely dependent on interactions present in the RGC's excitatory synaptic inputs.

**Figure supplement 2**. Rod → cone interactions and photoreceptor selectivity depend on mean luminance.

**Figure supplement 3**. Design of psychophysical experiments presented in *Figure 1*.

---

sensitivity of L-cone synaptic terminals and horizontal cells. We adjusted the strengths of rod and cone flashes until they elicited similar amplitude responses in ON RGCs (with resulting flash strengths within a factor of two of those used for probing rod-cone interactions in *Figure 1*). Then, in the same retinal slice, we measured voltage responses from either L-cone synaptic terminals (*Figure 2B*) or horizontal cells (*Figure 2C*) to the same flashes; horizontal cells receive direct synaptic input from cones and hence provide a direct measure of cone synaptic output (*Trumpler et al., 2008*) (*Figure 2A*). The ratios of the amplitudes of the rod-mediated to cone-mediated responses were ~8 times smaller in L-cone synaptic terminals and horizontal cells than in ON RGCs (*Figure 2D*). Thus, at these light levels rod-mediated signals appear to be routed through the dedicated rod bipolar circuitry until they reach the inner retina. Independent pharmacological manipulation of the rod and cone bipolar circuits provided further support for this conclusion (*Figure 2—figure supplement 1*). This situation differs considerably from mouse retina, where rod signals are transmitted in parallel through rod and cone (via rod-cone gap junctions) bipolar circuits at these light levels (*Grimes et al., 2014*; *Ke et al., 2014*).

Although rod-cone coupling appears to make a relatively modest contribution to the strength of the rod responses observed in the retinal output, such coupling could still account for rod → cone interactions—for example, if rod-mediated signals alter the gain of the cone output synapse. This did not appear to be the case: under conditions in which the RGC excitatory synaptic inputs exhibited strongly nonlinear responses, horizontal cells, AII amacrine cells, and cone bipolar cells all combined rod- and cone-mediated responses linearly—that is, the linear sum of the responses delivered individually matched the response to the flashes delivered sequentially (*Figure 2E*). The absence of nonlinear rod → cone interactions in circuit components upstream of the synapse between ON cone bipolar cells and RGCs, and presence of such interactions in the RGC excitatory synaptic inputs (*Figure 2F*), indicates that such interactions must occur at the cone bipolar output synapse itself. Because this synapse is shared by both the rod and cone bipolar circuits, it could function as a gate keeper, controlling which types of photoreceptor signals are transmitted to an ON parasol RGC.

What mechanisms mediate rod-cone interactions at the cone bipolar → RGC synapse? One possibility is that transmission of the response to the adapt flash depletes the pool of presynaptic vesicles, resulting in a depressed response to the test flash. This mechanism, however, cannot easily account for the greater strength of rod → cone interactions compared to cone → rod interactions (*Figure 1D,E*) since similar synaptic activation, regardless of origin, would be expected to produce similar levels of vesicle depletion. An alternative, but not exclusive, hypothesis is that kinetically-distinct rod- and cone-mediated signals sum linearly before passing through a common synaptic nonlinearity (i.e., the cone bipolar → ganglion cell synapse). Consistent with this hypothesis, the kinetics of the rod- and cone-mediated voltage responses of ON cone bipolar cells differed substantially (*Figure 3A*; similar differences were observed in AII amacrine responses): rod-mediated responses were biphasic, consisting of an initial depolarization (rightward blue arrow in *Figure 3B*) followed by a slow hyperpolarization (i.e., overshoot; leftward blue arrow in *Figure 3B*); the latter component depended on inhibition within the rod bipolar circuit (*Figure 3—figure supplement 1*). Cone-mediated responses, in comparison, were faster and substantially less biphasic (red arrows in *Figure 3B*). In this case, even if rod- and cone-mediated responses sum prior to the nonlinearity, the hyperpolarization associated with the rod-mediated

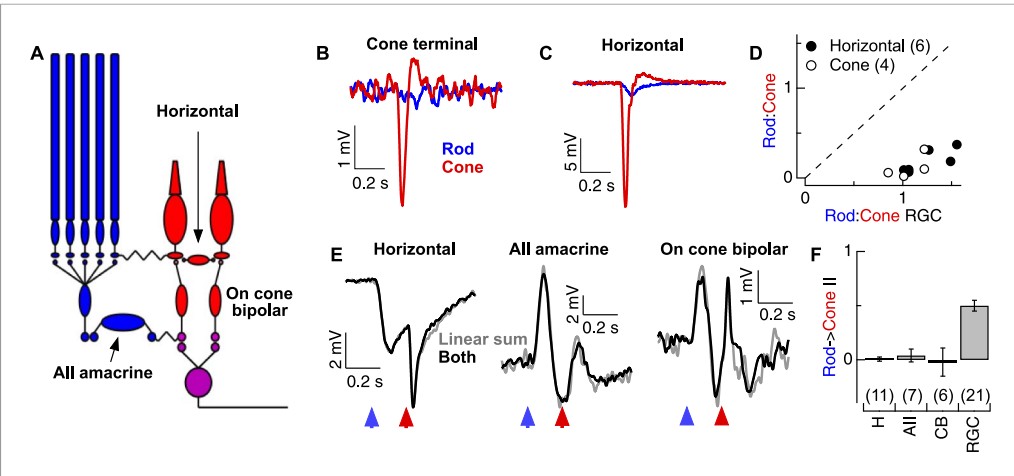

**Figure 2**. Rod and cone-mediated signals are largely independent upstream of the cone bipolar → RGC synapse. (**A**) Retinal circuit diagram. (**B–C**) Rod and cone flashes that produced near-equal amplitude responses in RGCs elicited strong cone responses and weak rod responses in the axon terminals of L-cones (**B**) and in horizontal cells (**C**). (**D**) Summary across cells. Ratios of the responses to rod- and cone-preferring stimuli in horizontal (H) and L-cone terminals plotted vs the response ratios in RGCs to the same flashes from the same slices. (**E**) Rod → cone interactions were absent in circuit elements upstream of the RGC: (left) horizontal cells, (middle) AII amacrine cells and (right) cone bipolar cells. (**F**) Mean rod → cone interaction indices for all horizontal, AII amacrine, cone bipolar and retinal ganglion cells (mean ± SEM, number of cells in parenthesis). Interaction indices for horizontal cells (p = 0.49), AIIs (p = 0.45) and cone bipolars (p = 0.49) did not differ significantly from 0, while those of RGCs did (p < 10$^{-7}$). All recordings except the RGCs reported in **F** (whole mount) from retinal slices.

The following figure supplement is available for figure 2:

**Figure supplement 1**. Pharmacological evidence that rod- and cone-mediated signals traverse the retina through largely distinct circuitry.

response could suppress the ability of a subsequent cone-mediated response to traverse the nonlinear synapse and produce a response in the ganglion cell. The smaller hyperpolarization of the cone-mediated response, however, would suppress a subsequent rod-mediated response less and hence result in a smaller nonlinear interaction.

The mechanistic hypothesis illustrated in *Figure 3B* predicts that rod → cone suppression should be maximal when a presynaptic test signal arrives during the peak of the presynaptic hyper-polarizing overshoot associated with the adapt flash response. To test this prediction, we examined rod → cone interactions present in the excitatory synaptic inputs to ON parasols across a range of time offsets (*Figure 3C*). Indeed, rod → cone suppression exhibited a time course consistent with that of the hyperpolarization in the rod-mediated ON cone bipolar voltage response (*Figure 3A,C, D*). Across cells, rod → cone suppression was maximal for time offsets between 0.1 and 0.3 s, with little or no suppression for longer or shorter offsets (*Figure 3D*). The behavior at time offsets <0.1 s is consistent with previous work showing that responses to simultaneously delivered rod and cone stimuli can be described by linear summation followed by saturation for strong stimuli (*Enroth-Cugell et al., 1977*; *Cao et al., 2010*).

Can linear summation followed by a synaptic nonlinearity quantitatively account for the measured rod → cone and cone → rod interactions? To answer this question, we characterized the relation between a time-varying stimulus (input) and the excitatory ganglion cell response (output) using a linear-nonlinear cascade model (LN model; see 'Materials and methods'), and then used this description to generate a parameter-free model for rod → cone and cone → rod flash interactions (*Figure 4B–E*). We derived the LN model components (i.e., the linear filter and static nonlinearity) for both rod and cone inputs using gaussian noise stimuli. Linear filters for rod inputs were slower and more biphasic than those for cone inputs (*Figure 4B*), consistent with the differences observed in ON cone bipolar voltage responses (*Figure 3A*). The nonlinearities derived from rod and cone inputs were similar (*Figure 4C*), consistent with a location in a shared element in the rod and cone circuits.

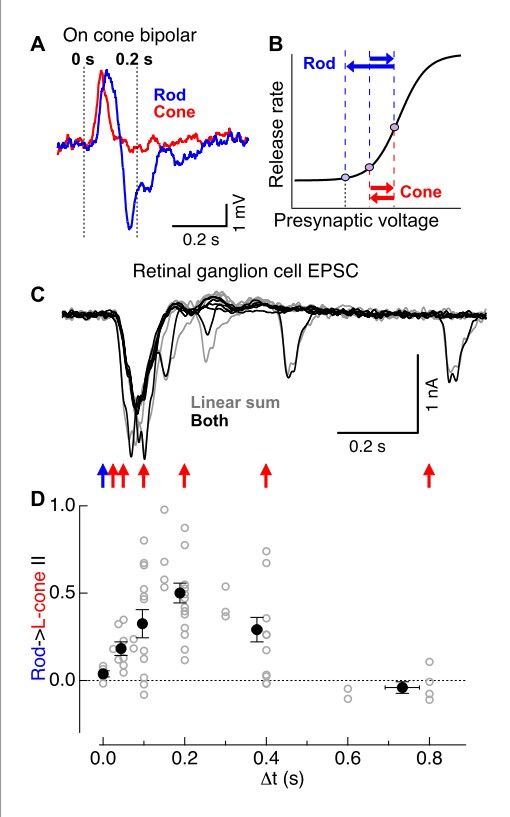

**Figure 3**. Rod → cone interactions exhibit a similar time course as the observed overshoot in cone bipolar cell voltage following a rod flash. (**A**) Cone bipolar voltage responses to rod and cone flashes exhibited distinct kinetics. (**B**) Illustration of impact of kinetic differences on cone bipolar synaptic output. Rightward blue and red arrows indicate depolarizing components of rod- and cone-mediated responses, while leftward arrows indicate hyperpolarizing components. (**C**) Time course of rod → cone interactions probed by delivering paired flashes across a range of temporal offsets. (**D**) Summary of time dependence of rod → cone interactions (individual cells in gray, population averages binned by Δt in black). Similar to the overshoot of the rod-mediated response in cone bipolar cells (**A**), rod → cone interactions were maximal when flashes were offset by 0.1–0.3 s. On cone bipolar cell recordings from retinal slices, and On parasol recordings from whole mount retina.

The following figure supplement is available for figure 3:

**Figure supplement 1**. Inhibition creates a rapid overshoot in rod-mediated signals at mean light levels where both rods and cones are active.

LN model components—the separate linear filters for rod and cone responses and a single common nonlinearity—were used to predict rod → cone and cone → rod flash interactions. First, we predicted responses to rod and cone flashes delivered independently. For example, for the rod flash response we scaled the amplitude of the rod linear filter so that its output, passed through the nonlinearity, matched the amplitude of the measured rod flash response. We followed the same procedure to scale the cone linear filter to match the measured cone flash response. We then predicted interactions between these flashes by offsetting the scaled linear filters in time to represent the time offset between flashes, linearly summing the two scaled filters, and passing the result through the common nonlinearity (*Figure 4D*).

Rod-cone interactions and LN model components were measured in the same cell so that the predicted (*Figure 4E*) and measured (*Figure 4F*) interactions could be directly compared. Across cells, the predicted and measured interaction indices were strongly correlated (*Figure 4G*). Both the predicted and measured interaction indices scaled with the degree of rectification in the nonlinearity as measured by the LN model (data not shown). The greater strength of rod → cone compared to cone → rod interactions in the LN model predictions depends directly on the rod linear filter being more biphasic than the cone filter; specifically, ~200 ms after the flash, the overshoot of the rod filter is maximally effective at suppressing the ability of the cone-mediated response to traverse the nonlinearity. Similarly, predicted interactions peaked for time offsets near 200 ms, as observed experimentally (*Figure 3D*). Thus a simple model based on linear summation of kinetically-distinct rod and cone responses followed by a rectifying nonlinearity can quantitatively predict the strength of interactions between rod- and cone-mediated flash responses, including the asymmetry in relative interaction strength.

## Discussion

The results here bear on the mechanistic basis of rod-cone interactions and more generally on parallel processing and computation in neural circuits. Our work leads to two broad conclusions. First, perceptual measures of rod-cone interactions and their dependence on light level are often interpreted in the context of the different routes that rod-mediated signals could take through the retina (reviewed in *Buck, 2004*, *2014*). Surprisingly, we found that the rod bipolar circuit remains the dominant route through which rod-mediated signals traverse the retina at light levels well above cone threshold. This lack of mixing of rod- and cone-mediated signals both constrains potential sites of interaction and provides ample opportunities for selective processing, for example to shape the kinetics of rod-mediated signals

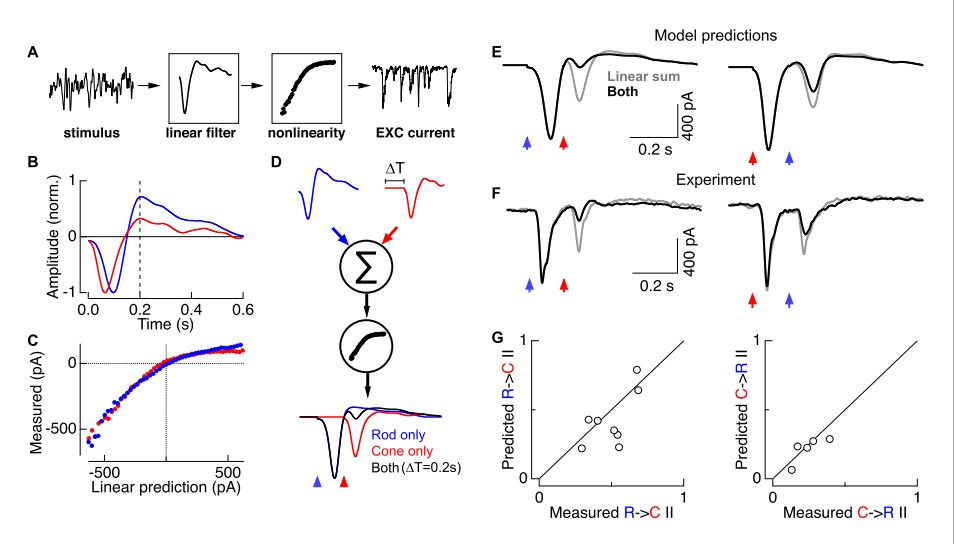

**Figure 4**. Linear summation followed by a rectifying nonlinearity can account for rod-cone interactions. (**A**) Linear-nonlinear (LN) model construction. A time-varying rod- or cone-preferring stimulus and the resulting RGC excitatory synaptic inputs were used to derive the linear filter and static nonlinearity that relate the stimulus to the response. (**B**) Normalized linear filters for rod and cone stimuli. (**C**) Nonlinearities for rod and cone stimuli. (**D**) Rod-cone interactions were predicted by summing scaled and temporally-offset (i.e., 0.2 s) rod- and cone-preferring filters and passing the result through a common nonlinearity (see 'Materials and methods'). (**E**) Predicted rod → cone (left) and cone → rod (right) interactions. (**F**) Measured rod → cone and cone → rod interactions. (**G**) Population data comparing predictions from the LN models to experimental observations. Each point represents a single cell in which LN model components and rod-cone interactions were measured. All recordings from whole mount retina.

distinctly from cone-mediated signals. Second, perceptual rod-cone interactions can be complex—for example, the asymmetric nonlinear interactions we investigate here. We show that these apparently complex interactions can be explained by a simple neural mechanism: linear summation of kinetically-distinct outputs from parallel circuits followed by a shared synaptic nonlinearity. This simple picture provides a key step towards understanding the mechanistic basis of the diverse rod-cone interactions that influence so many aspects of visual perception.

# Materials and methods

## Electrophysiology

Experiments were conducted on whole mount or slice (200 µm thick) preparations of primate retina as previously described (*Dunn et al., 2007*; *Trong and Rieke, 2008*). In brief, pieces of retina attached to the pigment epithelium were stored in ~32–34°C oxygenated (95% O$_2$/5% CO$_2$) Ames medium (Sigma, St Louis, MO) and dark-adapted for >1 hr. Pieces of retina were then isolated from the pigment epithelium under infrared illumination and either flattened onto polyL-lysine slides (whole mount) or embedded in agarose and sliced. Once under the microscope, tissue was perfused with Ames medium at a rate of ~8 ml/min. Non-human primate retina was obtained through the Tissue Distribution Program of the Regional Primate Research Center at the University of Washington.

Extracellular recordings from ON parasol retinal ganglion cells used ~3 MΩ electrodes containing Ames medium. Voltage-clamp whole-cell recordings were conducted with electrodes (RGC: 2–3 MΩ, AII: 5–6 MΩ) containing (in mM): 105 Cs methanesulfonate, 10 TEA-Cl, 20 HEPES, 10 EGTA, 2 QX-314, 5 Mg-ATP, 0.5 Tris-GTP and 0.1 Alexa (488, 555 or 750) hydrazide (~280 mOsm; pH ~7.3 with CsOH). Current-clamp whole-cell recordings were conducted with electrodes (AII: 5–6 MΩ, CB: 8–12 MΩ, HC: 5–6 MΩ, L-cone terminals: 10–12 MΩ) containing (in mM): 123 K-aspartate, 10 KCl, 10 HEPES, 1 MgCl$_2$, 1 CaCl$_2$, 2 EGTA, 4 Mg-ATP, 0.5 Tris-GTP and 0.1 Alexa (488, 555 or 750) hydrazide (~280 mOsm; pH ~7.2 with KOH). Cone terminals were identified by following axons from cone cell bodies to the margin of the outer plexiform layer. In initial experiments, cell types were confirmed

by fluorescence imaging following recording. NBQX (10 µM; Tocris) was added to the perfusion solution as indicated in *Figure 2—figure supplement 1*. To isolate excitatory or inhibitory synaptic input, cells were held at the estimated reversal potential for inhibitory or excitatory input of ∼−60 mV and ∼+10 mV. These voltages were adjusted for each cell to maximize isolation. Dynamic clamp procedures followed those described previously (*Murphy and Rieke, 2006*). Absolute voltage values have not been corrected for liquid junction potentials (K$^+$-based = −10.8 mV; Cs$^+$-based = −8.5 mV).

## Psychophysics

The dichoptic apparatus consisted of two 60 Hz LCD computer monitors (1920 × 1200 Dell, model U2412M) controlled by a Mac mini computer running Psychtoolbox for Matlab (*Brainard, 1997*; *Pelli, 1997*). The observer's left and right eyes viewed separate monitors. NDF0.6, 'Bright pink', and 'Scarlet' gel filters (Rosco E-colour, Stamford, CT) were mounted to the front of each monitor to control luminance and suppress the transmission of wavelengths between 500–600 nm, thus improving photoreceptor selectivity for the red and blue phosphors. The apparatus was uniquely aligned for each observer's session. Monitor translation (coarse adjustments) and image translation (fine adjustments) were made sequentially under the direction of the observer to maximize overlap of the images in the two eyes and invoke binocular fusion. After acceptable fusion was achieved, observers were dark-adapted for 10–15 min before beginning the tasks. Observers occasionally lost fusion within a session; in these cases the session was exited and the alignment procedure was repeated before returning to the task.

Two versions of the matching task produced qualitatively similar results. In the first version (v1), observers repeatedly adjusted the brightness of the bottom test flash and then manually indicated a perceptual match. In the second version (v2), observers adjusted the intensity of the bottom test flash until they had reversed the direction of their adjustments 6 times, at which point they were automatically advanced to the next trial (*Figure 1—figure supplement 3*). In both versions of the task the adjustment increments were reduced by 1/3 after each crossing until reaching the minimum permissible intensity adjustment set by the discrete monitor intensity values. For each session, observers ran 10–16 trials under each condition (e.g., rod → cone [short], rod → cone [long], cone → rod [short]); monocular and binocular trials were randomly interleaved. Matches were averaged across trials within each session before calculating the interaction index.

In Task v1, we excluded data from trials in which observers achieved fewer than 2 crossings and from any conditions with 3 or fewer measurements/trials. In Task v2, trial matches were calculated by averaging the midpoints between contiguous pairs of crossing values, with each midpoint weighted by the inverse of the step size between that pair of crossings. In both tasks, retinal interactions were inferred by comparing the perceptual matches obtained when adapt and test flashes were delivered to the same vs separate eyes. Interactions arising from separate eye delivery must occur in higher brain regions with access to information from both eyes (e.g., cortical interactions) whereas interactions arising from same eye delivery would also include retinal interactions. We assumed that retinal and cortical gain factors were multiplicative and occurred in series. Thus suppressive interactions attributable to the retina were taken as the ratio of the perceptual sensitivities for 1 vs 2 eye delivery. Interaction indices were calculated for each session before averaging across sessions. The strength of perceptual interactions attributable to the retina were defined using an interaction index analogous to that used for the physiological experiments:

$$II = 1 - \frac{S_{1eye}}{S_{2eye}},$$

where $S_{1eye}$ is the sensitivity ratio for single eye delivery (i.e., intensity ratio of the perceptual match) and $S_{2eye}$ is the sensitivity ratio for separate eye delivery. This index is 0 if interactions for one and two eye stimuli have identical strengths and 1 if the adapt flash completely suppresses perception of the test flash for one but not two eye stimuli.

Each observer conducted at least 1 training session before data was included for analysis. These training sessions helped observers (1) become comfortable with binocular fusion of images from distinct monitors, (2) improve their fixation while flashes are delivered to the peripheral retina and (3) become comfortable with the user interface (e.g., entering responses, transitions between conditions). Visual perception experiments were conducted according the human subject guidelines laid forth by the University of Washington.

## Visual stimuli

For electrophysiology experiments, full field illumination (diameter: 500–560 μm) was delivered to the preparation through a customized condenser from blue (peak power at 460 nm) or red (peak power at 640 nm) LEDs (*Figure 1—figure supplement 2*). Light intensities (photons/μm$^2$/s) were converted to photoisomerization rates (R*/photoreceptor/s) using the estimated collecting area of rods and cones (1 and 0.37 μm$^2$, respectively), the stimulus (i.e., LED or monitor) emission spectra and the photoreceptor absorption spectra (*Baylor et al., 1984*, *1987*). In *Figures 1, 3* the blue LED provided a constant illumination of ∼20 R*/rod/s. In *Figure 4* the blue and red LEDs produced a mean of ∼20 R*/rod/s and ∼200 R*/L-cone/s. Rod- and cone- preferring flashes were 10 ms in duration.

For visual perception experiments, red (peak power at 640 nm) and blue (peak power at 444 nm) spots of ∼2° were presented for 16 ms at ∼10° eccentricity in human observers. The filtered backlight of the monitor produced a mean luminance of ∼1 R*/rod/s. Flash isomerization estimates for all experiments are presented in *Figure 1—figure supplement 2*.

## LN model

We used a linear-nonlinear (LN) model to test whether summation of rod- and cone-mediated responses followed by a shared synaptic nonlinearity could account for rod-cone interactions. The linear filter (L) and nonlinearity (N) were estimated using measured excitatory synaptic inputs to On parasol ganglion cells in response to 50% contrast Gaussian noise (0–60 Hz bandwidth). Stimulus wavelength and intensity were chosen to emphasize rod- or cone-mediated responses (see *Figure 1—figure supplement 2*). Model components were estimated using standard approaches (*Rieke, 1997*; *Chichilnisky, 2001*; see *Figure 4A*). In brief, correlating the response with the stimulus estimated the linear filter (*Figure 4B*). Comparing the filter output (i.e., the stimulus convolved with the filter) with the measured response provided an estimate of the nonlinearity (*Figure 4C*). Ganglion cell responses to flashes were then predicted by scaling the linear filter according to the flash strength and passing the scaled filter through the nonlinearity (*Figure 4D*).

## Analysis

Nonlinear interactions cause the neural response to the test flash to differ in the presence or absence of the adapt flash. We quantified these interactions in our electrophysiology data by using an interaction index, defined as:

$$II = 1 - \frac{R_{pair}}{R_{single}},$$

where $R_{single}$ is the integral of the response to the test flash delivered alone and $R_{pair}$ is the integral of the response to the test flash when preceded by the adapt flash (i.e., the response to the paired flashes minus the response to the adapt flash alone).

Rod signals recorded in AII amacrine cells were biphasic under the luminance conditions tested here (i.e., ∼20 R*/rod/s; *Figure 3—figure supplement 1*). We quantified this property using a biphasic index, defined as:

$$BI = \frac{A_H}{A_H + A_D},$$

where $A_D$ is the maximum amplitude of the depolarization and $A_H$ is the maximum amplitude of the hyperpolarization in the first 400 ms of the baseline corrected response.

Electrophysiology example traces presented throughout the figures represent the average of 5–20 raw responses to the same stimuli. All data are presented as mean ± SEM and two-tailed paired Student's t-tests were used to test significance.

## Acknowledgements

We thank Steve Buck, Greg Horwitz, EJ Chichilnisky, Michael Do, Andreas Liu and Greg Bryman for critical reading of the manuscript, and Michael Ahlquist, Mark Cafaro and Shellee Cunnington for excellent technical assistance. Support provided by HHMI and the NIH (EY11850 and 5R90DA033461).

# Additional information

## Funding

| Funder | Grant reference | Author |
|---|---|---|
| Howard Hughes Medical Institute (HHMI) | | Fred Rieke |
| National Institutes of Health (NIH) | EY11850 | Fred Rieke |
| National Institutes of Health (NIH) | 5R90DA033461 | Mathew T Summers |

The funders had no role in study design, data collection and interpretation, or the decision to submit the work for publication.

## Author contributions

WNG, LRG, MTS, FR, Conception and design, Acquisition of data, Analysis and interpretation of data, Drafting or revising the article

## Ethics

Human subjects: The experimental protocol was approved by the Institutional Review Board of the University of Washington (protocol 16934) and was in accordance with the Declaration of Helsinki. All subjects gave informed consent in writing before participating in the experiment.

Animal experimentation: We obtained primate retinas (*Macaca fascicularis*, *Macaca nemestrina* and *Macaca mulatta* of either sex, ages 3–19 years) through the Tissue Distribution Program of the Regional Primate Research Center. All protocols were approved by the Institutional Animal Care and Use Committee at the University of Washington (protocol 4140-01).

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
