## [Decision Letter]

Thank you for submitting your work entitled “Retinal interactions between day and night visual pathways: mechanism to perception” for peer review at *eLife*. Your submission has been favorably evaluated by Timothy Behrens (Senior editor) and two reviewers, one of whom is a member of our Board of Reviewing Editors.

The following individuals responsible for the peer review of your submission have agreed to reveal their identity: Ronald L Calabrese (Reviewing editor) and Joshua Singer (peer reviewer). The reviewers have discussed the reviews with one another and the Reviewing editor has drafted this decision to help you prepare a revised submission.

Summary:

The authors present an interesting electrophysiological, psychophysical, and modeling analysis of rod cone interactions in the primate retina. Psychophysical experiments in humans show that rod-cone interactions are asymmetric with rod responses suppressing subsequent cone responses much more strongly than vice versa. Similarly cone ON parasol ganglion cells show the asymmetry in retinal whole-mounts and slices. Careful analysis shows that the site of the asymmetry is best understood as linear summation at the ON cone bipolar cell synapse followed by the non-linearity in the synapse of the bipolar cell onto the ON parasol ganglion cell. The rod input here arises from the AII amacrine cell and is biphasic accounting for the suppressive interval after a rod flash. LNL modeling corroborates this analysis. The tie in between the biophysics and the slice physiology gives a special appeal to the results and the careful nature of the analysis gives confidence to the conclusions. The paper should be a wide interest in the visual community and also to other sensory physiologists because it serves as an example of how inputs from parallel processing pathways are combined for functional output.

Essential revisions:

The reviewer concerns are in agreement and are summarized below. The authors should answer each query and modify the presentation accordingly.

1) The mechanisms that give rise to the biphasic input to the ON parasol ganglion cells from the AII amacrine cell was not explored/explained. This issue should at least be addressed in the Discussion, perhaps including what is known in the mouse.

2) Figure 1: it's interesting that ganglion cell spiking is decreased for a period following cone stimulation but not rod stimulation. Is this a consistent observation? What do the authors make of it?

3) What accounts for the hyperpolarizing overshoot associated with the rod response? The authors must have some synaptic mechanism in mind. In the same vein, what might AII amacrine and ON cone bipolar responses to paired cone flashes look like? Would there be no hyperpolarization (as in Figure 2)? I wonder whether repeated cone stimulation could elicit a hyperpolarization in the ON cone bipolar.

4) Suppose two rod responses were evoked? Presumably they would interact nonlinearly if the biphasic rod driven response is determining the properties of the paired-pulse interactions. And two cone responses would sum linearly? As an aside, is linear summation of cone responses part of the reason the temporal precision of cone pathways is higher than that of rod pathways?

5) The citation of [14] is confusing. The conclusion of this paper is that rod signals move through the dedicated rod bipolar cell circuit at relatively high light levels-just like the supplement to Figure 2, the Ke et al. paper shows that NBQX blocks rod evoked responses in AIIs—and in this respect, the situation in the primate may not differ considerably from that of the mouse, as the authors write (paragraph starting “Previous studies provide…”). Or do the authors mean that Ke et al. show that ganglion cell responses to flashes that activate rods are suppressed by backgrounds (reproducing earlier results from the Rieke lab)?

---

## [Author Response]

*1) The mechanisms that give rise to the biphasic input to the ON parasol ganglion cells from the AII amacrine cell was not explored/explained. This issue should at least be addressed in the Discussion, perhaps including what is known in the mouse*.

We are currently exploring the kinetics of rod signals as they traverse the primary rod pathway. Initially we thought the biphasic responses were generated through a combination of mechanisms, and that this issue was better served as part of a separate study. Our data now point to a simpler picture in which inhibitory feedback, likely to the rod bipolar synaptic terminal, is the primary contributor. We have included this data as a supplement to Figure 3 (Figure 3—figure supplement 1) and mention it in the Results. Details for those experiments are now presented in the Methods and associated figure legend.

*2)*
Figure 1*: it's interesting that ganglion cell spiking is decreased for a period following cone stimulation but not rod stimulation. Is this a consistent observation? What do the authors make of it?*

We do not consistently see larger suppression of firing following cone stimulation compared to rod stimulation (indeed both rod and cone responses of Figure 1 show clear suppression, with the suppression of firing following rod stimulation lasting considerably longer than that following cone stimulation). Spike responses often exhibit periods of suppressed activity following the initial burst, regardless of flash origin. Rectification in the synaptic inputs, as well as low maintained firing rates (which often hit zero 200–400 ms following a flash), minimize the overshoots in ganglion cell spike responses and can cause them to look similar for rod- and cone-mediated responses. In some cases (such as the gray trace in Figure 1) the suppression reflects a combination of a slow suppression following rod activation and a briefer suppression following cone activation.

*3) What accounts for the hyperpolarizing overshoot associated with the rod response? The authors must have some synaptic mechanism in mind. In the same vein, what might AII amacrine and ON cone bipolar responses to paired cone flashes look like? Would there be no hyperpolarization (as in*
Figure 2*)? I wonder whether repeated cone stimulation could elicit a hyperpolarization in the ON cone bipolar*.

As stated in response to comment #1, we have now included new data which speaks to the origin of the overshoot in the rod responses observed in AIIs and CBs. These experiments suggest that feedback inhibition to the rod bipolar terminal accounts for the majority of the rapid component of the overshoot. Cone-mediated responses in AII amacrine cells look very similar to those in ON cone bipolar cells; in particular, such responses do not show a prominent overshoot. We have not measured responses to paired cone flashes in AIIs or ON cone bipolar cells. Relevant here though, linear filters describing the cone-mediated excitatory synaptic inputs to ganglion cells are substantially less biphasic than filters for rod-mediated inputs (Figure 4). These filters are calculated during continuous inputs, so the lack of overshoot of the cone-mediated responses to brief flashes appears reasonably general and extends to stimuli involving prolonged cone activation

4) Suppose two rod responses were evoked? Presumably they would interact nonlinearly if the biphasic rod driven response is determining the properties of the paired-pulse interactions. And two cone responses would sum linearly? As an aside, is linear summation of cone responses part of the reason the temporal precision of cone pathways is higher than that of rod pathways?

We are currently working to compare the mechanisms and functional impact of paired-flash interactions within the rod vs. cone bipolar circuits, including rod-rod and cone-cone interactions. These experiments focus on the role of synaptic mechanisms like depression and inhibition, and how those mechanisms differ across circuits. We felt that those more mechanistically focused and more complicated experiments distracted from the focus of the present paper on the connection between retinal signaling and perception. As the reviewer intuits, however, paired flash interactions in On Parasols are strongest for pairs of rod flashes (strongly nonlinear) and weakest for pairs of cone flashes (mostly linear).

We hesitate to draw a connection between linear summation and temporal precision because nonlinearities could help or hinder temporal precision; e.g. thresholding nonlinearities could improve temporal precision by removing spurious response fluctuations.

*5) The citation of*
[14]
*is confusing. The conclusion of this paper is that rod signals move through the dedicated rod bipolar cell circuit at relatively high light levels-just like the supplement to*
Figure 2*, the Ke et al. paper shows that NBQX blocks rod evoked responses in AIIs—and in this respect, the situation in the primate may not differ considerably from that of the mouse, as the authors write (paragraph starting* “*Previous studies provide…*”*). Or do the authors mean that Ke et al. show that ganglion cell responses to flashes that activate rods are suppressed by backgrounds (reproducing earlier results from the Rieke lab)?*

Thanks, that point was indeed confusing as worded. The key distinction we want to draw is that rod signals in mouse traverse the retina through both rod and cone bipolar circuits, while in primate the rod bipolar circuit appears to dominate. Thus, in mouse retina, both [14] and [13] show that rod responses in AII amacrine cells are only partially blocked by NBQX at luminance levels above ∼10R*/ rod/s. The responses remaining in the presence of NBQX are interpreted as coming through the rod->cone->cone bipolar pathway (similar to Trexler et al. 2005). This contrasts to our observations in primate where the rod-evoked flash response is eliminated by NBQX at a mean background of ∼20R*/rod/s. The statement now reads: ‘This situation differs considerably from mouse retina, where rod signals are simultaneously transmitted in parallel through rod and cone (via rod-cone gap junctions) bipolar circuits at these light levels (13; 14)’.